# Long-Term Symptoms after Mild Coronavirus Disease in Healthy Healthcare Professionals: A 12-Month Prospective Cohort Study

**DOI:** 10.3390/ijerph20021483

**Published:** 2023-01-13

**Authors:** Grazielle Rosa da Costa e Silva, Winny Éveny Alves Moura, Kamila Cardoso dos Santos, Davi Oliveira Gomes, Gabriela Nolasco Bandeira, Rafael Alves Guimarães, Claci Fátima Weirich Rosso, Gabriela Silvério Bazilio, Vanessa Rafaela Milhomem Cruz Leite, Karlla Antonieta Amorim Caetano, Megmar Aparecida dos Santos Carneiro, Sheila Araújo Teles

**Affiliations:** 1Faculty of Nursing, Federal University of Goiás, Goiânia 74605-080, GO, Brazil; 2Institute of Biological Science, Federal University of Goiás, Goiânia 74690-900, GO, Brazil; 3Institute of Tropical Pathology and Public Health, Federal University of Goiás, Goiânia 74605-050, GO, Brazil

**Keywords:** COVID-19, healthcare workers, long COVID, SARS-CoV-2 infection, symptoms

## Abstract

The coronavirus disease 2019 (COVID-19) pandemic has changed the course of human history and killed millions of people worldwide. Its long-term consequences remain uncertain. This study aimed to describe the short- and long-term symptoms of COVID-19 among individuals in Goiás, central Brazil, who experienced acute mild or non-symptomatic SARS-CoV-2 infection during the first wave of the pandemic. This prospective cohort study included 110 healthcare workers, 18 safety workers, and 19 administrative support workers, who were followed up for 12 months after the onset of COVID-19. Most participants were healthy adult female healthcare professionals. At the onset of infection, the major symptoms were headache, myalgia, nasal congestion, cough, coryza, anosmia, ageusia, sore throat, fatigue, diarrhea, and dyspnea. Furthermore, 20.3% of the participants had three or more COVID-19 symptoms that persisted for at least 12 months. These included coryza, congestion, hair loss, sore throat, headache, myalgia, cough, memory loss, anosmia, and fatigue. This study revealed a high prevalence of persistent symptoms of COVID-19 in healthy individuals from central Brazil, which may present an additional burden on healthcare services. Further studies are required to investigate the sequelae of COVID-19 over periods greater than 12 months.

## 1. Introduction

Severe acute respiratory syndrome coronavirus 2 (SARS-CoV-2) causes coronavirus disease 2019 (COVID-19). Since its discovery in December 2019, this RNA virus has affected more than 620 million people and caused more than 6.5 million deaths worldwide [1]. The first cases in Brazil were detected in February 2020 and quickly spread to all regions of the country. Currently, Brazil is the country with the fifth-highest number of COVID-19 cases [1]. This virus has evolved over time, and new variants with increased transmissibility and ability to evade the immune response have emerged. Consequently, many researchers have focused on the epidemiology of the SARS-CoV-2 infection and have presented insights into the clinical features of its acute phase [2,3,4].

SARS-CoV-2 is transmitted via air droplets and mainly affects the respiratory system. Although most infected individuals remain asymptomatic or exhibit mild short-term symptoms [5,6], 2% may develop a severe multisystemic disease that affects the nervous and hematopoietic systems as well as the heart, kidney, liver, and muscles [6,7].

Knowledge about COVID-19 has expanded continuously since the start of the pandemic [8,9]. COVID-19 may persist into the post-acute phase [6]. However, accumulating information on the duration of the long-term effects of COVID-19 is dependent on the duration of the observation period.

Most studies on COVID-19 have reported its long-term sequelae among individuals with severe disease [10]. However, data on the long-term sequelae in healthy adults who have experienced acute mild or non-symptomatic COVID-19 are still scarce. Understanding these symptoms is crucial as they may have an impact on quality of life and productivity.

According to the Centers for Disease Control and Prevention (CDC), long-term COVID-19 represents “a wide range of new, returning symptoms or ongoing health problems that people experience after being infected with the virus that causes COVID-19”. These symptoms can be continuous or intermittent. They occur among people who experienced all clinical forms of COVID-19, although more frequently among those who experienced severe disease [11].

The long COVID consequences can be multisystemic and include cardiovascular, neurological, pulmonary, dermatological, and psychological alterations. Although the exact mechanism responsible for these complications is still unknown, it is believed that the direct viral tissue damage and the entry receptor for SARS-CoV-2, angiotensin-converting enzyme 2 (ACE2), which is largely expressed in several cells of the body, play a role in the long COVID presentation [12].

During the first wave of the COVID-19 pandemic in Brazil, the Federal University of Goiás implemented a multidisciplinary project named “COVID-19 Tent” to screen healthcare and safety workers. This was later extended to monitoring healthy individuals who had recovered from COVID-19 to identify the short-term and long-term disease sequelae.

This study aimed to identify the short- and long-term symptoms of COVID-19 among people who experienced acute mild or non-symptomatic SARS-CoV-2 infection during the first wave of the pandemic in Goiás, central Brazil.

## 2. Materials and Methods

Healthcare and safety workers who reported one or more COVID-19 symptoms were screened in the “COVID-19 Tent”. Between July and August 2020, 408 workers tested positive for SARS-CoV-2. These individuals were invited to participate in this prospective cohort study, and 147 (36%) agreed to do so.

After obtaining informed consent, participants were interviewed about their socio-demographic characteristics; they were asked to indicate the presence or absence of short-term symptoms of COVID-19 by providing “yes” or “no” responses to a predefined symptom checklist that included the following: fever, fatigue, myalgia, arthralgia, headache, anosmia, ageusia, chest pain, respiratory discomfort, palpitations, cough, coryza, nasal congestion, sore throat, diarrhea, nausea, vomiting, abdominal pain, and conjunctivitis. They were free to report any additional symptoms that were not on the list and the medications they took for treatment. The post-COVID condition usually starts approximately 1 month after the onset of COVID-19 with symptoms, can persist for a year or longer, and cannot be explained by an alternative diagnosis [13].

Detection of SARS-CoV-2 infection was performed using probes from the 2019-nCoV kit (Integrated DNA Technologies, San Diego, CA, USA), which detects the sN1 and sN2 regions of SARS-CoV-2 as well as the RP encoding the human RNase P protein, according to the protocol by the CDC. After duplicate testing, samples were reported as inconclusive if they presented amplification of only one viral target (N1 or N2) or amplification of two viral targets after cycle 37 of reverse transcription quantitative real-time PCR (RT-qPCR) amplification.

### 2.1. Statistical Analysis

Data were analyzed using Stata version 13.0 (StataCorp., College Station, TX, USA). Distributions of frequencies (proportion, mean, std deviation, median, and interquartile range) were calculated. The Student’s t-test and chi-squared test or Fisher’s exact test were used to determine the statistical significance of comparisons between continuous variables and categorical variables, respectively. Statistical significance was set at *p* < 0.05.

### 2.2. Ethical Statement

This study was conducted in accordance with the Declaration of Helsinki and was approved by the Institutional Ethics Committee for Human Research of the Clinical Hospital of the Federal University of Goiás (protocol number: 31542620.7.0000.5078). Written informed consent was obtained from all participants.

## 3. Results

The mean age of the participants was 38.4 years. The majority were female (63.9%), 32.7% were white, 53.1% were of mixed race, and 74.8% were employed as healthcare workers. The most frequently reported chronic diseases were hypertension (38.8%), asthma/bronchitis (23.1%), chronic cardiac disease (9.5%), diabetes mellitus (4.1%), and chronic gastrointestinal disease (1.4%) (Table 1).

At diagnosis, the participants reported the following symptoms (Table 2): headache (76.9%), cough (70.7%), coryza (63.9%), sore throat (62.6%), myalgia (61.9%), nasal congestion (59.2%), fatigue (51.7%), fever (55.1%), ageusia (46.3%), anosmia (43.5%), diarrhea (40.1%), respiratory discomfort (26.5%), and nausea (26.5%). Only five participants reported that they had no COVID-19 symptoms.

The most commonly used medicines for managing the acute symptoms of COVID-19 were azithromycin (43.5%), ivermectin (38.8%), and hydroxychloroquine (10.9%). The participants were followed for a median of 290 days (IQR: 57–372 days). The median interval between the onset of symptoms and a positive RT-PCR test was 4 days (IQR: 3–5 days). None of the participants were hospitalized with COVID-19.

At the 1-month follow-up (median: 27 days, IQR: 24–30 days), 128 participants were evaluated, of whom 26.6%, 19.5%, and 24.2% reported three or more symptoms, two symptoms, and one symptom, respectively (Figure 1). These symptoms included headache (28%), myalgia (24.2%), nasal congestion (21.1%), cough (21.9%), coryza (18.8%), anosmia (16.4%), ageusia (14.8%), sore throat (12.5%), fatigue (10.9%), diarrhea (7.8%), dyspnea (3.1%), and nausea (2.3%) (Figure 2).

At the 3-month follow-up (median: 94 days, IQR 88–101 days), 108 participants were evaluated, of whom 11.1%, 14.8%, and 22.2% reported three or more symptoms, two symptoms, and one symptom, respectively. These symptoms included headache (21.3%), coryza (10.2%), nasal congestion (13%), myalgia (13%), anosmia (8.4%), fatigue (6.5%), ageusia (6.5%), cough (5.6%), sore throat (4.6%), and memory loss (3.7%).

At the 6-month follow-up (median: 183 days; IQR, 151–194 days), 91 participants were evaluated, of whom 14.3%, 14.3%, and 19.8% reported three or more symptoms, two symptoms, and one symptom, respectively. These symptoms included headache (22.3%), myalgia (16.5%), nasal congestion (13.2%), cough (12.8%), coryza (12.1%), fatigue (11.7%), sore throat (9.9%), memory loss (6.6%), anosmia (7.7%), hair loss (6.6%), and ageusia (4.3%).

At the 9-month follow-up (median: 276 days, IQR: 268–288 days), 90 participants were evaluated, of whom 34.4%, 24.4%, and 14.4% reported three or more symptoms, two, and one symptom. These symptoms included headache (31.1%), coryza (31.1%), hair loss (26.7%), memory loss (26.7%), nasal congestion (22.2%), myalgia (18.9%), cough (15.6%), fatigue (13.3%), sore throat (12.2%), anosmia (7.8%), and ageusia (4.4%).

At the 12-month follow-up (median: 375 days, IQR: 364–390 days), 69 participants were evaluated, of whom 20.3%, 26.1%, and 34.8% reported three or more symptoms, two, and one symptom, respectively. These symptoms included coryza (36.2%), nasal congestion (29%), hair loss (20.9%), sore throat (17.4%), headache (14.5%), myalgia (14.5%), cough (14.5%), memory loss (13%), anosmia (8.7%), fatigue (7.2%), and ageusia (1.4%).

Of all the symptoms, only reports of fatigue were proportionally less frequent at 1 month after onset between individuals followed up for 12 months (4.3%) compared to those who were not (18.6%) (*p* < 0.05). Furthermore, there were no significant differences regarding sex and occupation between individuals followed up at 12 months and those followed up at 1 month after symptom onset.

There were no reports of a SARS-CoV-2 reinfection during the 12 months of follow-up. After COVID-19 vaccines became available to healthcare workers in Brazil, 95 participants who were still enrolled in the study reported receiving the Oxford/Covishield Vaccine (Fiocruz and AstraZeneca), CoronaVac vaccine (Butantan), Janssen vaccine (Janssem-Cilag), or Comirnaty vaccine (Pfizer–Wyeth). 

## 4. Discussion

To our knowledge, the present study is the first to identify the persistence of mild symptoms for at least 12 months after the diagnosis of COVID-19 among healthy individuals with mild or asymptomatic disease in Brazil.

The short-term symptoms in our cohort were similar to those reported by Rodriguez-Morales et al. [14] in a systematic review of symptoms reported during the first wave of the pandemic. Although SARS-CoV-2 variants have emerged throughout the pandemic, the associated COVID-19 symptoms have been similar, differing only in frequency and intensity [15].

A few previous studies [16,17] have reported female sex, the presence of more than five initial symptoms, and being a healthcare professional as risk factors for long-term COVID-19. Notably, our cohort mainly comprised females and healthcare professionals; furthermore, 81% of the patients had at least five initial symptoms.

In our cohort, most symptoms reported at the onset of COVID-19 were still present at the one-month follow-up. This finding is consistent with the findings of a systematic review by Lopez-Leon et al. [18], which identified more than 50 long-term COVID-19 symptoms 15–110 days after SARS-CoV-2 infection, of which the majority of these were similar to the symptoms experienced during the acute phase of COVID-19.

Headache, myalgia, and respiratory symptoms were the most frequently reported initial symptoms at diagnosis and the most frequently reported persistent symptoms among those who were followed up for at least 12 months. These long-term symptoms have been reported mainly among individuals hospitalized due to COVID-19 [19,20].

In Madrid, a study was performed on 615 patients who were hospitalized due to COVID-19. It revealed that after a mean period of 7.3 months after discharge, 40.8% and 42.5% of the patients reported one or two symptoms and three or more post-COVID symptoms, respectively, of which headache, fatigue, and respiratory symptoms were the most common symptoms reported [21]. Similarly, in an Italian study, 143 patients were assessed at a mean of 2 months after the onset of COVID-19 symptoms, of whom 32% and 55% reported one or two symptoms, and three or more symptoms, respectively. Fatigue, dyspnea, and joint pain were the most frequently reported symptoms [22].

Some hypothesized mechanisms in post-acute COVID-19 pathophysiology include autoimmunity [23], inflammatory and metabolic changes in the parenchyma and supporting structures during the initial infection, and sequelae mediated by hospitalization interventions [19]. Our findings confirm that long-term COVID-19 symptoms are not dependent on the severity of the acute disease.

Immune-mediated tissue damage in COVID-19 involves both cellular and humoral responses, but the mechanisms underlying immunity to SARS-CoV-2 and protection against reinfection or final viral clearance are unknown. In addition, the reason some patients experience long-term symptoms after COVID-19, whereas others do not, is unclear. Some host factors that influence the outcome of viral infection may play a role, for example, genetic susceptibility, age, infecting dose, route of infection, induction of anti-inflammatory cells and proteins, presence of comorbidities, and cross-reactive immunity due to previous exposure to related viruses. It is still unknown whether SARS-CoV-2 can cause substantial tissue damage leading to a chronic form of the disease, similar to the chronic lesions observed during convalescent stages of infections by other viruses (such as human immunodeficiency virus, hepatitis C virus, hepatitis B virus, and some herpesviruses).

Studies point to the pre-existing conditions as potentiators of long-term COVID-19. The reactivation of the Epstein–Barr virus and the human immunodeficiency virus (HIV) were associated with fatigue symptoms and neurocognitive changes, in addition to symptoms of post-traumatic stress and functional impairment. The immune system and organ tissue damage may contribute to long-term COVID-19 [24,25,26].

The results of the present study are consistent with current scientific knowledge of other coronaviruses, including those causing severe acute respiratory syndrome (SARS) and Middle East respiratory syndrome, both of which share clinical characteristics (including post-acute phase symptoms) with COVID-19. Studies on survivors of SARS have shown that lung abnormalities months after infection may induce an atypical mast cell response and increased interleukin-6 and angiotensin-converting enzyme 2 levels in the nervous system, resulting in headaches [27,28,29].

The respiratory system is the primary target of SARS-CoV-2, which influences the breakdown of endothelial and epithelial barriers, allowing monocytes and neutrophils to invade the alveolar space [30]. Thereafter, SARS-CoV-2 reaches the olfactory bulb, regions of the cortex, midbrain, and basal ganglia, leading to respiratory problems such as nasal congestion and coryza [31].

It is noteworthy that some patients reported memory and hair loss some months after their COVID-19 diagnosis. Hair loss in COVID-19 may be associated with the action of proinflammatory cytokines, such as interleukin 1β, interleukin 6, tumor necrosis factor α, and interferon-γ [32]. This hair loss could be attributed to telogen effluvium, which is a condition characterized by diffuse hair loss following an important systemic stressor or an infection. It is caused by premature follicular transitions from the active growth phase (anagen) to the resting phase (telogen). It is a self-limiting condition that lasts for approximately three months and can cause significant emotional distress.

Memory loss from COVID-19 is thought to be a consequence of the structural and metabolic neurological alterations caused by the infection [17]. The following brain changes may be related to memory loss: global decrease in the gray matter volume; decreased gray matter volumes in the left Rolandic operculum, right cingulate, both hippocampi, and left Heschl’s gyrus; and decreased cerebral blood flow [33].

Most participants in our study consumed antiparasitic drugs and/or antibiotics for COVID-19. During the period in which cohort members were recruited, many doctors in Brazil prescribed these medicines for acute COVID-19, despite the lack of evidence of effectiveness [34]. In addition, the Brazilian president encouraged the use of these medications for the “early” treatment of COVID-19 [35]. However, subsequent studies have not found any evidence of the effectiveness of these drugs for COVID-19 treatment [36]. Further, studies even indicate the existence of life-threatening side effects in the gastric, neurological, and mainly the cardiac system due to the alteration in the QT segment in cardiac waves [37,38].

The limitations of our study include the loss of participants to follow-up over time because they dropped out of the study despite attempts made to keep them in the cohort. Therefore, survival bias cannot be excluded, and the proportion of symptoms should be considered with caution. However, it is noteworthy that at 1 month after onset, the proportion of all but one symptom (fatigue) was similar between individuals who followed up for 12 months compared to those who were not. In addition, although the self-reported symptoms were consistent with those documented in the literature, there may have been a few inaccuracies. Despite these limitations, our study provides the first account of the long-term symptoms of COVID-19 in central Brazil over a period of 12 months.

## 5. Conclusions

This study revealed that three or more symptoms of COVID-19 can persist for up to a year in healthy people with mild COVID-19. Upper tract respiratory symptoms were the most common. Further studies are required to investigate the sequelae of COVID-19 over periods greater than 12 months.

## Figures and Tables

**Figure 1 ijerph-20-01483-f001:**
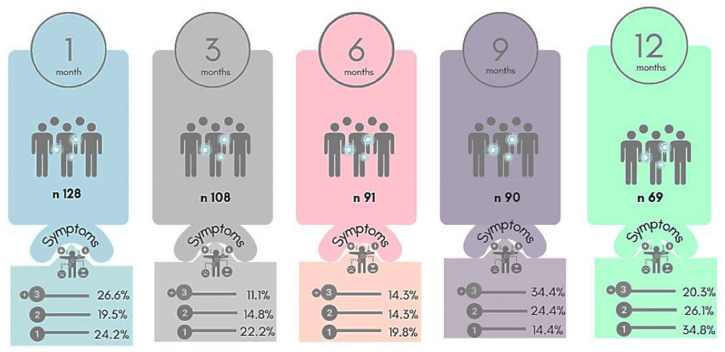
Number of symptoms during the follow-up. Licensed under CC BY 4.0. To view a copy of this license, visit http://creativecommons.org/licenses/by/4.0/ (accessed on 7 November 2022).

**Figure 2 ijerph-20-01483-f002:**
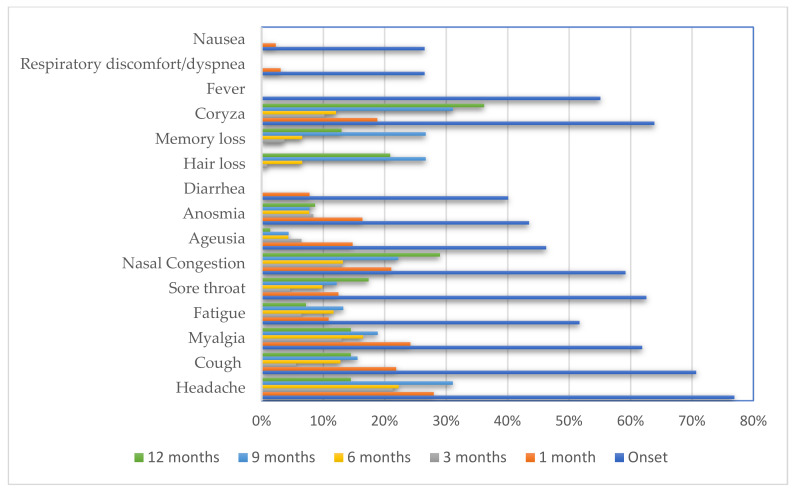
Persistence of self-reported symptoms of COVID-19 according to the time since onset of COVID-19. COVID-19; coronavirus disease 2019.

**Table 1 ijerph-20-01483-t001:** Characteristics of 147 healthcare and safety workers diagnosed with coronavirus disease 2019 (COVID-19) in central Brazil.

Characteristic	Value	%
Age, mean years (SD)	38.4 (10.2)	
Sex		
Female	94	63.9
Male	53	36.1
Race		
White	48	32.7
Mixed	78	53.1
Black	17	11.6
Asian	4	2.6
Occupation		
Healthcare worker	110	74.8
Safety worker	18	12.3
Others	19	12.9
Comorbidities		
Hypertension	57	38.8
Asthma/bronchitis	34	23.1
Chronic cardiac disease	14	9.5
Chronic renal disease	8	5.4
Diabetes mellitus	6	4.1
Chronic gastrointestinal disease	2	1.4
Chronic liver disease	1	0.7
Human immunodeficiency virus infection	1	0.7
Medicines used		
Azithromycin	64	43.5
Ivermectin	57	38.8
Hydroxychloroquine	16	10.9
Analgesics, antipyretics, and vitamins	75	51.0
Other medicine	58	39.5
Duration of follow-up, median (IQR), days	290 (57–372)	
Duration of SARS-CoV-2 RNA positivity, median (IQR), days	14 (9–32)	

COVID-19: coronavirus disease 2019; IQR: interquartile range; SARS-CoV-2: severe acute respiratory syndrome coronavirus 2; SD: standard deviation; Others: administrative support occupations.

**Table 2 ijerph-20-01483-t002:** Self-reported initial symptoms of coronavirus disease 2019 (COVID-19).

Characteristic	N	%
Headache	113	76.9
Cough	104	70.7
Coryza	94	63.9
Sore throat	92	62.6
Myalgia	91	61.9
Nasal congestion	87	59.2
Fatigue	76	51.7
Fever	81	55.1
Ageusia	68	46.3
Anosmia	64	43.5
Diarrhea	59	40.1
Respiratory discomfort/dyspnea	39	26.5
Nausea	39	26.5

COVID-19: coronavirus disease 2019.

## Data Availability

Database is available as a Appendix A.

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
