# Peer review of "Long-Term Symptoms after Mild Coronavirus Disease in Healthy Healthcare Professionals: A 12-Month Prospective Cohort Study"

_ijerph, 2023, doi:10.3390/ijerph20021483_

Round 1
Reviewer 1 Report
The authors present the interesting results of long-term symptoms of mild COVID-19. These are important statistical results for investigators who are interested in studying the symptoms of COVID-19. The manuscript will be ready for publication if the major concerns can be addressed.
1. How do the authors confirm that these symptoms exist before the patient has COVID-19?
2. It would be great if the authors can provide the specific types of vaccine
3. One of the authors' finding is that the long-term COVID-19 symptoms are not dependent on the severity of the acute disease. However, it is contradicted to some of the results in the following references. My concern is the authors studied the mild cases and conclude this conclusion, which is bias.
CDC: https://www.cdc.gov/nchs/pressroom/nchs_press_releases/2022/20220622.htm
WHO: https://jamanetwork.com/journals/jama/fullarticle/2797443
Variants: https://www.thelancet.com/journals/lancet/article/PIIS0140-6736%2822%2900941-2/fulltext
Vaccine: https://www.thelancet.com/journals/eclinm/article/PIIS2589-5370%2822%2900354-6/fulltext
https://jamanetwork.com/journals/jama/fullarticle/2794072
4. Need more references to support on lines 32 to 34 "Consequently, several researchers have focused on the epidemiology of a SARS-CoV-2 ..."
5. In the introduction, more discussions about long-term symptoms for all situations. The introduction is too short.
6. Missing "%" in line 99
7. There are 147 participants. Thus it would be important to present how many participants in the current follow-up attend the previous one.
8. Update the figure 1's caption. Describe those 3, 2, 1 in the middle part.
Reviewer 2 Report
This question is of particular concern to family doctors worldwide! It is particularly relevant for employees in the health sector that employees suffer from long-term symptoms and may be absent as a result.
For me, the following points are missing from the discussion:
- Bias due to lost to follow-up? Have the patients not come back because they have no symptoms? Is it not possible to follow them up after all?
- Influence of the medication side effect, which was used here without evidence?
- Influence of secondary diagnoses or pre-existing conditions - such as COPD or depression? There are also various studies on this
It would be good to include these points in the discussion and possibly also discuss other large studies in the COVID follow-up.
https://pubmed.ncbi.nlm.nih.gov/36379135/
Round 2
Reviewer 1 Report
The authors have addressed all my concerns. This manuscript is ready for publication after some minor improvements.
1. To leverage the manuscript to be more interesting to the audience, the authors should extend the second and third paragraphs of the introduction.
2. Figure 1. I am guessing the y-label is the percentage?
3. The authors should expand their conclusion as well.
4. a typo in 244
Author Response
Por favor, verifique o anexo.
